# The Ocular Surface and the Anterior Segment of the Eye in the Pseudoexfoliation Syndrome: A Comprehensive Review

**DOI:** 10.3390/ijms26020532

**Published:** 2025-01-10

**Authors:** Maya Natasha Thomas, Piotr Skopiński, Harry Roberts, Małgorzata Woronkowicz

**Affiliations:** 1NDDH, Royal Devon University Healthcare NHS Foundation Trust, Barnstaple EX31 4JB, UK; mayanatasha.thomas@nhs.net; 2Department of Ophthalmology, SPKSO Ophthalmic University Hospital, Medical University of Warsaw, 00-576 Warsaw, Poland; pskopin@wp.pl; 3Department of Histology and Embryology, Medical University of Warsaw, 02-004 Warsaw, Poland; 4West of England Eye Unit, Royal Devon University Healthcare NHS Foundation Trust, Exeter EX2 5DW, UK; harry.roberts@nhs.net; 5Faculty of Health and Life Science, University of Exeter Medical School, Exeter EX1 2HZ, UK; 6Moorfields Eye Hospital NHS Foundation Trust, London EC1V 2PD, UK

**Keywords:** pseudoexfoliation syndrome, pseudoexfoliation glaucoma, ocular surface, conjunctiva, Tenon’s capsule, sclera, cornea, iris, ciliary body, zonules, aqueous humour, trabecular meshwork, lens

## Abstract

Pseudoexfoliation syndrome (PXS) is an age-related fibrillopathy where fibrillar exfoliation material accumulates and deposits in ocular and extra-ocular tissue. Within the eye, this substance accumulates on the ocular surface and in the anterior segment of the eye, impacting ocular structures such as the conjunctiva, Tenon’s capsule, sclera, cornea, iris, ciliary body, trabecular meshwork, and lens. This review aims to collate the current literature on how each anatomical part of the eye is affected by PXS, with a strong focus on molecular changes. We also summarise the current understanding of the key genetic factors influencing the development of PXS.

## 1. Introduction

Pseudoexfoliation syndrome (PXS) is described as an age-related fibrillopathy where white fibrillar exfoliation material accumulates and deposits in ocular and extra-ocular tissue [1,2]. Glaucoma is the leading cause of irreversible blindness globally. PXS often leads to pseudoexfoliation glaucoma (PXG), which accounts for up to 25–70% of all glaucoma cases worldwide, making it the leading cause of open-angle glaucoma [2].

The prevalence of PXS increases with age, with most cases occurring in individuals over the age of 60 [1,3]. As populations around the world age, the burden of PXS is expected to rise, particularly in countries with large elderly populations. Therefore, understanding the global epidemiology of PXS is critical for assessing its public health impact and informing preventive measures, especially in ageing populations. The prevalence of PXS is greater in women than men [3]. This could be related to women having on average higher life expectancies than men.

Pseudoexfoliative material is composed of various extracellular matrix (ECM) components such as fibrillin-1, elastin, and fibronectin, as well as cross-linked glycoproteins [3,4,5]. It is insoluble and contains lysosomal enzymes that degrade the tissue it surrounds [3]. The term “pseudoexfoliation” was coined to distinguish it from true exfoliation syndrome, which describes the rare delamination of superficial and deep layers of the anterior lens capsule secondary to heat, infra-red damage, trauma, or intra-ocular inflammation [6].

A growing body of evidence shows that PXS deposits can be found extra-ocularly, around blood vessels and in connective tissue around the heart, lung, and brain [7]. It has been associated with Alzheimer’s disease, cerebral atrophy, ischaemic heart disease, renal artery stenosis, aortic aneurysms, and systemic hypertension, among others [1,8,9]. In the eye, pseudoexfoliation material has been demonstrated on the ocular surface and in the anterior segment of the eye [3,10,11,12].

The *lysyl oxidase-like 1* (*LOXL1*) gene, encoding the *LOXL1* enzyme, is the most well-studied and well-established genetic contributor to PXS; however, its role is not yet fully understood [1,4,13,14,15,16,17,18]. The *LOXL1* protein has been shown to be one of the prime components of the pseudoexfoliative material found in PXS and pseudoexfoliative glaucoma (PXG) [15,16]. The *lysyl oxidase* enzymes are a family of five enzymes known for their role in cross-linking and stabilising collagen and elastin [1,16]. *LOXL1* regulates tropoelastin, a precursor of elastin, and plays a role in the formation and remodelling of elastin [16]. Variants in the *LOXL1* gene, particularly single nucleotide polymorphisms (SNPs), have shown strong associations with PXS in various populations [1,19,20,21]. The three most significant variants are *rs1048661*, *rs3825942*, and *rs2165241*, which have been found to be genetic risk factors for Icelandic, Swedish, American, Australian, and European cohorts [1,15,19]. One study showed that *rs4886776*, *rs1048661*, and *rs2165241* SNPs exhibited allelic reversal depending on ethnic group (Japanese and Chinese populations) [20]. Interestingly, the *rs3825942* SNP is protective against PXS in South African populations [21]. Changes in LOXL1 expression are also seen in ageing populations and other elastin connective tissue disorders [1].

While LOXL1 is the primary gene associated with PXS, several others have been identified in genome-wide association studies (GWASs), though they have not shown the same consistency across all populations. The genes *CACNA1A*, *CYP39A1*, *POMP*, *TMEM136*, *AGPAT1*, *RBMS3*, *SOD2*, *ALDH1A1*, and *SEMA6A* have also been found to play a role in developing PXS [13,14,19,20,22,23,24].

*CYP39A1* is responsible for metabolising cholesterol by converting 24S-hydroxycholesterol (24S-OHC) into downstream intermediates and variations in *CYP39A1* have been strongly associated with PXS [25,26,27]. *CYP39A1* expression is reduced in PXS with corresponding increases in 24S-OHC and 24S-OHC/cholesterol ratios in the aqueous humour and in serum, which increases oxidative stress and induces free radical generation [25,26]. It is suggested that *CYP39A1* deficiency and high concentrations of 24S-OHC are involved in the pathogenesis of PXS [25,26,27]. The ocular tissues of PXS patients demonstrate significantly lower expression levels of genes associated with cholesterol synthesis (e.g., *HMGCR*, *SREBF2*), efflux (e.g., *ABCA1*, *ABCG1*), transport (e.g., *APOB*, *APOC1*), and regulation (e.g., *LXRA/B*, *RXRA*) compared to control tissues [25,26]. One study that analysed 35 PXS patients with the *CYP39A1* mutation found that the former group had a significantly increased risk of PXG, and more severe glaucoma and blindness compared to *CYP39A1*-negative patients [28]. This suggests further research into the role of impaired cholesterol metabolism in PXS pathogenesis could be explored.

Large scale genome-wide association studies have linked the *Calcium Voltage-Gated Channel Subunit Alpha 1* (*CACNA1A*) gene to PXS with an odds ratio of 1.16, suggesting its role in the pathogenesis of PXS [20]. This association has been established within the populations of 18 countries, including Japan, Argentina, Poland, India, and South Africa [20]. Interestingly, no significant difference has between found in *CACNA1* mRNA expression within various ocular tissues in PXS and control groups [29]. Studies on PXS eyes have shown high calcium levels associated with PXS fibrils with fibrillin, a component of PXS aggregates, relying on calcium for stability [30]. It has been hypothesised that altered calcium channel function could disrupt calcium levels, potentially promoting the formation of PXS aggregates [31].

Other factors such as UV exposure, oxidative stress, folate deficiency, and overconsumption of coffee have also been found to be associated with developing PXS [4,14,32,33,34]. The mechanisms behind these associations are not fully understood. In a multivariable analysis, the risk of PXS has been found to increase by 1.5% for each additional sunny day per year. Moreover, the use of sunglasses has been shown to protect against the development of PXS [32]. A positive association between consuming over 500 mg/day of caffeine and PXS and PXG has been demonstrated [33]; however, conflicting data have emerged [32]. A 25% reduction in risk of developing PXG has been associated with high dietary folate [34]. Moreover, elevated serum homocysteine levels have been linked to PXS and PXG [34]. It has also been demonstrated that excessive coffee consumption and low dietary folate consumption are associated with raised homocysteine levels [34]. Further research is needed to understand how both those factors impact the development of PXS.

## 2. Methodology

Our literature search was conducted using PubMed, Google Scholar, and Web of Science online databases. Keywords such as “pseudoexfoliation” and “pseudoexfoliation syndrome” in combination with terms referring to ocular surface and anterior segment structures were used to identify relevant articles. Boolean operators were employed to make searches more precise. Citations were manually searched to identify additional articles that were relevant.

### 2.1. The Effect of Pseudoexfoliation on Ocular Structures

Several structures of the ocular surface and anterior segment of the eye (Figure 1) can be affected by pseudoexfoliative material (Figure 2).

### 2.2. Conjunctiva

The conjunctiva is a thin, transparent, mucous membrane with roles including ocular immunity, homeostasis, and maintaining the tear film [35]. It contains numerous goblet cells producing mucin, which is an integral part of the tear film [35,36].

Pseudoexfoliation deposits have been found in the conjunctiva [37]. Pseudoexfoliative material is thought to impair tear film integrity and reduce tear film production [38,39]. PXG patients have been shown to have a more unstable tear film than PXS patients [38,39]. Electron microscopy studies have demonstrated impaired goblet cell activity within the conjunctiva of PXS patients [37]. While the number of goblet cells was not reduced compared to controls, their distribution, size, appearance, and structural integrity were abnormal, resulting in their dysfunction and subsequently more common dry eye disease in PXS patients [37,39]. Moreover, individuals affected by PXS are also more likely to develop conjunctivochalasis characterised by loose redundant conjunctival folds [40].

### 2.3. Tenon’s Capsule

Under the conjunctiva sits Tenon’s capsule, a thin and elastic fascial sheath that surrounds the eyeball [41] and, along with the conjunctiva, terminates at the cornea [35,36,41].

PXG patients have abnormal cellular arrangement of lysosomes, autophagosomes, and microtubules in Tenon’s capsule fibroblasts [42]. Fibroblasts have a broad range of functions including creating and remodelling the extracellular matrix, acting as progenitors for various mesenchymal cells, and signalling areas of tissue damage to stem cells to position tissue repair [43].

Want et al. found that Tenon’s fibroblasts from PXS patients showed lysosomal, endosomal, and mitochondrial dysfunction [43]. They applied stress to cells by creating a starvation state and used immunostaining to identify Tenon’s fibroblasts in PXS patients and controls. Abnormal distribution of lysosomes and endosomes was observed in response to cellular stress. LC3, a marker of autophagy, was found at a 3.9 times higher concentration in PXS fibroblasts than controls, which indicated reduced autophagosome clearance. Impaired cellular autophagy in turn resulted in reduced mitophagy. This abnormal accumulation of non-functioning mitochondria is thought to reduce the cell’s response to oxidative stress. Fibulin-5, another elastin regulator protein, was expressed 8–10 times more in fibroblasts from PXS patients. The authors postulate that fibulin-5 could be an important marker of dysfunction in these cells. They also described that Tenon’s capsules derived from PXS patients were 1.38 times larger than age-matched controls, with fibroblasts in eyes with PXS not showing the normal parallel alignment typical for the control group. In addition, *LOXL1* expression within Tenon’s fibroblasts has been shown to increase in response to oxidative stress, UV exposure, hypoxia, and transforming growth factor beta-1 (TGF)-β1 [4].

### 2.4. Sclera

The sclera is a white fibrous tissue that encloses most of the eye, providing structure and protection to it [44]. It is composed of collagen, elastin, glycoproteins, and proteoglycans arranged irregularly, contributing to its opacity [44].

Little is known about how PXS affects the sclera. Anterior scleral thickness has been shown to be similar in PXS and control groups [45]. The lamina cribrosa and the peripapillary sclera are part of the posterior sclera, which surrounds the optic nerve. A study of three donor eyes with PXS found that lamina cribrosa was over 40% less stiff and peripapillary sclera nine times stiffer in PXS patients compared to the control group [46]. More research is needed to understand the significance of these data and to uncover more about how the sclera is altered by pseudoexfoliative material.

### 2.5. Cornea

The cornea is a transparent, avascular, optically clear structure made of five layers: the corneal epithelium, Bowmann’s layer, the corneal stroma, Descemet’s membrane, and the corneal endothelium [47], as seen in Figure 3. Clinically, the Sampolesi line, which appears as increased pigmentation anterior to Schwalbe’s line, can be seen on gonioscopy and is pathognomonic in individuals with PXS and pigment dispersion syndrome [48].

Several studies investigated the impact of PXS on the histological structure of the cornea [49,50,51,52,53,54,55,56,57,58,59]. Patients with PXS have reduced basal corneal epithelial as well as keratocyte stromal cell counts [50,52,53].

Pseudoexfoliative material and phagocytosed melanin have also been observed within the corneal endothelium [51]. PXS is associated with increased rates of endothelial polymegathism and pleomorphism [49,50,51,52]. Moreover, endothelial cell density (ECD) can be reduced in PXS [53]. One study has found a significant reduction in ECD in PXG eyes when compared to eyes with primary open angle glaucoma and suggested that raised intra-ocular pressure and PXS damage the endothelium via different mechanisms [54]. Light microscopy studies have demonstrated pseudoexfoliative material on the corneal endothelium and embedded within Descemet’s membrane, resulting in posterior outpouchings [60]. Results suggested that it was created locally by endothelial cells generating damaged pockets within the endothelium, exposing Descemet’s membrane and becoming embedded in it as endothelial cells regenerated. PXS initially does not affect corneal transparency; however, as it becomes more severe, endothelial decompensation and corneal damage could occur [51]. Performing phacoemulsification early in patients with PXS was found to limit corneal damage and reduce the risk of complications [56].

PXS is also associated with impaired corneal sub-basal nerve plexus and nerve fibre density [52,53,55]. Impairment of the corneal sub-basal nerve plexus correlates with reduced corneal sensation and is more marked in PXS patients with hyperreflective endothelial deposits than those without [53,55].

Various studies have analysed how PXS affects corneal cell density with varying results. A study by Oltulu et al. employed confocal microscopy in 27 patients with PXS and found a reduction in all corneal cell densities apart from corneal epithelial cells [55]. Yilmaz et al. reported the opposite when they studied 94 eyes with PXS, finding that corneal density increased with worsening severity of PXS, with more pronounced changes in the peripheral cornea [57]. A study that analysed Pentacam HR imaging on 31 patients with unilateral PXS found no statistically significant difference in corneal densitometry between the affected and unaffected eyes [58].

Yazgan et al. reported a statistically significant reduction in mean central corneal thickness (CCT) and corneal hysteresis in 73 PXS and PXG patients [59]. Mean CCT was found to be 509 ± 36 μm, 525.5 ± 35 μm, and 546.3 ± 28 μm in PXG, PXS, and healthy controls, respectively. Moreover, corneal resistance factor in PXS was also significantly reduced at 7.9 ± 1.6 mmHg compared to 10.3 ± 0.7 mmHg in the control group. In contrast, Pradhan et al. found no statistical difference in CCT or other corneal biomechanical parameters within their cohort, suggesting that different results were attributed to the effect of glaucoma medications on corneal biomechanical readings [61].

### 2.6. Iris

The iris controls the amount of light entering through the pupillary aperture by constricting and dilating [62]. It consists of the anterior and posterior iris epithelium, the iris stroma, the sphincter pupillae, and dilator muscle [62].

In PXS, white flecks of pseudoexfoliative material can be seen on the anterior and posterior surface of the iris [11,15]. Marked degeneration of the posterior iris epithelium results in focal membrane rupture and release of melanin granules [63,64]. This leads to the classic “moth-eaten” appearance of the pupillary margin, poor pupillary dilatation, iris transillumination defects and dispersion of melanin granules upon pharmacological dilation, which can be observed on clinical examination [10,63,64]. In addition, pseudoexfoliative material can damage the iris stromal vessels, typically accumulating first in the adventitia and involving endothelial cells of vessels, which can result in avascular blood vessel walls termed “ghost vessels” [18,64]. Consequently, micro-neovascularisation and collapse of iris stromal vessel lumen can develop, which can cause anterior chamber hypoxia [64,65,66].

A histopathological study of 33 eyes with PXS found surface cell membrane excavations with PXS fibres on various iris cells, suggesting local PXS production [65]. Moreover, pigment epithelial cell layers showed signs of focal disintegration with pseudoexfoliative material demonstrated on the apical aspects of the epithelial cells. Posterior synechiae (iridocapsular adhesions) can develop when pseudoexfoliative material causes the iris pigment epithelium to adhere to the anterior lens capsule and decreases iris mobility and elasticity [67].

The human natural killer-1 (HNK-1) epitope is a sulphated trisaccharide that plays a key role in many cell interactions including cell recognition, adhesion, and migration [12]. A study on six cadaveric eyes with pseudoexfoliation found that the pseudoexfoliative material on the posterior iris, ciliary body, anterior lens capsule, zonular fibres, and uveal trabecular meshwork showed strong HNK-1 antibody reactivity [12]. In contrast, PXS material in the iris stroma and juxtacanalicular trabecular meshwork was characterised by weak reactivity. This may suggest that the HNK-1 epitope could play a role in the adhesion of pseudoexfoliative deposits to intra-ocular surfaces.

Clusterin, also known as Apolipoprotein J, is a glycoprotein that is involved in various cellular processes throughout every cell in the body [68,69,70]. It acts as a molecular chaperone to inhibit the aggregation of misfolded proteins [68] and has been demonstrated to be a major component of pseudoexfoliation material [69,70,71,72,73]. Clusterin mRNA expression is reduced in the iris, lens epithelium, and ciliary processes of PXS eyes [5,68,71]. It is thought that clusterin deficiency within intra-ocular tissue results in the development of pseudoexfloliative deposits in the eye [68,71]. Notably, TGF-β stimulates clusterin synthesis in various cells and is significantly raised in the aqueous humour in PXS [5].

Structurally, PXS irises are noted to have a larger iris volume than in primary open angle glaucoma (POAG) [74]. Anterior segment Scheimpflug image studies of 60 patients with PXS reported that in all quadrants, the minimum lens–iris distance was shorter compared to controls [75]. An optical coherence tomography (OCT) study of the anterior chamber found that the pupils of PXS patients dilated 21% less than controls [76]. Moreover, the average midperipheral iris thickness was 7.8% thinner than in the control group. In a retrospective study of 511 patients with PXS, affected eyes had significantly shallower anterior chambers (2.92 ± 0.43 mm vs. 3.08 ± 0.41 mm) and larger lens thickness (4.78 ± 0.43 mm vs. 4.61 ± 0.43 mm) [10]. Pupillary diameter was also found to be significantly smaller in PXS eyes compared to non-PXS eyes (5.8 ± 1.1 mm vs. 6.9 ± 0.99 mm). Moreover, a higher incidence of floppy iris syndrome (IFIS), zonunolysis, and intra-operative pupil fixation was noted in PXS patients compared to non-PXS patients.

### 2.7. Ciliary Body, Zonules and Aqueous Humour

The ciliary body is a ring-shaped structure within the eye that has ridges called ciliary processes [77]. They secrete aqueous humour and are the attachment for zonules—a system of thread-like fibres that hold the lens of the eye centred and alter its shape to enable the eye to change focus [62,77].

PXS is thought to cause destruction of the ciliary epithelial basement membrane and subsequently degradation of the ciliary epithelium and the ciliary muscle [78]. *LOXL1* positive pseudoexfoliative material has been demonstrated on the ciliary body surface and on the ciliary muscle [18]. In addition, TGF-β1 mRNA and TGF-β1 protein levels have been found to be raised in the non-pigmented epithelium of the ciliary body of PXS eyes [79]. High levels of TGF-β1 are thought to impact wound healing after glaucoma surgery and influence the development of a secondary cataract, which is seen more commonly in patients with PXS [79].

In normal eyes, clusterin mRNA is found in high concentrations in the ciliary process epithelium [68,71]. In PXS, clusterin mRNA expression in the non-pigmented ciliary epithelium is significantly reduced [71]. Clusterin plays a role as a molecular chaperone and its deficiency is thought to promote aggregation and deposition of pseudoexfoliative material [71].

Pseudoexfoliative deposits on zonules and the zonular lamella are thought to induce zonular weakness and consequently lens subluxation [9,63]. A large retrospective Malaysian study on 12,992 eyes reported that PXS patients were 2.68 times more likely to develop a complication during cataract surgery [80]. Intra-operatively, zonular weakness can cause complications such as zonular dehiscence, vitreous loss, and most importantly, posterior capsule rupture [80,81]. Postoperatively, lens capsular bag phimosis has been reported in PXS eyes, which can further loosen zonular fibres and cause late in-the-bag lens dislocations [81].

The aqueous humour is produced by the non-pigmented epithelium of the ciliary body and bathes the anterior segment, providing important nutrients and maintaining optical clarity for clear vision [77]. It leaves the anterior chamber via the trabecular meshwork and drains through 25–30 collector channels to join the venous system via the deep scleral plexus [78].

Several studies analysed the composition of aqueous humour in eyes with PXS (Table 1). Pro-inflammatory cytokines and chemokines, such as interleukin (IL)-6, IL-8, TGF-β1, and tumour necrosis factor-alpha (TNF-α), have been found in elevated concentrations in PXS-affected eyes [82,83,84]. Inflammatory signalling may accelerate pseudoexfoliative material deposition and ECM instability, particularly in the trabecular meshwork, where inflammation contributes to impaired aqueous humour outflow and elevated intra-ocular pressure (IOP) [84]. Aqueous humour concentrations of IL-6 and IL-8 have been found to be increased threefold in the early stages of PXS when compared to controls [84]. Interestingly, levels of IL-6 and IL-8 in late stage PXS and PXG have not significantly differed from the control group [84].

In response to oxidative stress, IL-6 has been shown to induce expression of TGF-β1, which is a crucial cytokine in the immune cell development and regulation of the immune response [85]. Patients with PXS have significantly raised levels of TGF-β1 in their aqueous humour, which is key in the creation of pseudoexfoliative fibrillar deposits [5,79,86].

TNF-α is a proinflammatory cytokine affecting the body’s inflammatory response [85]. It is produced by activated macrophages, T-lymphocytes, and natural killer cells [85]. TNF-α binds to cellular receptors, triggering the release of numerous inflammatory molecules. The aqueous humour level of TNF-α has been found to be significantly increased in patients with early and late PXS, as well as PXG, compared to the controls [83]. Moreover, blood serum levels of TNF-α have also been demonstrated to be significantly higher in PXG patients [87].

Studies have shown raised levels of oxidative stress markers and reduced levels of antioxidants [14,83,88]. It is thought that oxidative stress is a driver of PXS [79,88,89]. Several reports have shown raised levels of Complement factor 3, antithrombin III, Kininogen-1, protein-carbonyl, vitamin D-binding protein, and 8-hydroxy-2′-deoxyguanosine [14,83,88,89,90]. PXS subjects have also been found to have a higher concentration of asymmetric dimethyl arginine (ADMA), which is known to increase oxidative stress and trigger cell apoptosis [88,91]. Interestingly, raised levels of ADMA have been reported in cardiovascular and metabolic diseases [88,91]. Moreover, the aqueous humour of PXG patients has shown a reduction in various antioxidants including selenium and glutathione [92,93].

Products of lipid peroxidation within the aqueous humour have been demonstrated to be raised in PXS patients compared to controls [88,89]. Lipid peroxidation occurs when free radicals react with unsaturated fats [94]. Gartanganis et al. found raised levels of glutathione disulfide and thiobarbituric acid reactive substance within the aqueous humour of PXS patients, implying high levels of oxidative stress [93]. A lipidomic analysis of aqueous humour in PXS, PXG, and POAG reported a lower total lipid content compared to controls [95].

The incidence of PXS increases markedly with age, indicating that age-related cellular senescence plays a role in PXS pathogenesis [1,96,97]. Levels of Klotho, a protein implicated in the ageing process, has been found to be reduced in PXS, PXG, and POAG [98,99]. Aqueous humour and serum concentrations of vascular endothelial growth factor have been noted to be elevated in patients with PXS, possibly indicating an ischaemic element to its pathogenesis [100]. A study on 25 patients with PXS found serum fetuin-A levels raised compared to control subjects but the significance of this finding is still not yet known [101].

**Table 1 ijms-26-00532-t001:** Aqueous humour components that have been found to be upregulated or downregulated in PXS.

Raised Concentration in Aqueous Humour	Refs.
IL-6	[84]
IL-8	[84]
TGF-β1	[83,85,86]
TNF-α	[87]
Protein-carbonyl	[90]
Complement factor 3	[88]
Antithrombin III	[88]
Kininogen-1	[88]
Vitamin D-binding protein	[88]
8-hydroxy-2′-deoxyguanosine	[89]
ADMA	[91]
Selenium	[92]
Glutathione	[88,93]
Thiobarbituric acid	[93]
VEGF	[100]
Fetuin-A	[101]
**Lowered concentration in aqueous humour**	
Total lipid content	[95]
Klotho	[98,99]

### 2.8. Trabecular Meshwork

The TM has three layers: the inner uveal meshwork, the middle corneoscleral meshwork, and the outer juxtacanalicular meshwork, as shown in Figure 4 [102]. The trabecular outflow pathway is the main one for aqueous humour to drain out of the eye [102]. Via this pathway, aqueous flows through the trabecular meshwork (TM), Schlemm’s canal, and the collector channels, and enter the aqueous veins [35,102].

Pseudoexfoliative material can be found on the trabecular meshwork and downstream of the trabecular outflow pathway i.e., Schlemm’s canal, the collector channels, aqueous, and episcleral veins [18,64,103]. It causes degenerative changes to Schlemm’s canal and surrounding tissues, which contributes to raised intra-ocular pressure [104]. Ultrastructural data suggest that pseudoexfoliative material in the inner uveal meshwork of the TM originates elsewhere in the anterior segment, whereas material in the juxtacanalicular TM seems to be produced locally by endothelial and connective tissue cells [105]. This is supported by how the juxtacanalicular region of the TM undergoes notable fibrotic changes in PXS [2,74] with electron microscopy data of PXG eyes showing trabecular cells to have a thickened plasma membrane, increased number of mitochondria and secretory organelles, as well as organised fibrils on the surface, as seen in Figure 5 [2]. ECM build-up, altered cytoskeletal structure, and increased cellular death are also involved in the increased resistance to aqueous outflow [86]. In vitro studies found that TM cells undergo epithelial–mesenchymal transition following chronic exposure to TGF-β1, which is raised in PXS [86]. This was confirmed by increased levels of inflammatory markers such as α-SMA, vimentin, IL-6, and IL-8 [86].

The ECM of the TM is composed of elastin and collagen complexes crosslinked by LOXL1 [2]. LOXL1 is detected in all ocular tissues but has the highest ocular expression within the TM [17]. Studies on mice with *LOXL1* deficiency demonstrated abnormal elastin fibres and the formation of pseudoexfoliative material similar to one found in PXS and PXG [2]. This suggests that LOXL1 deficiency results in abnormal elastin deposition, which resembles PXS.

Pseudoexfoliation is classified into PXS and PXG [74]. Individuals with exfoliative materials without raised intra-ocular pressures or nerve damage are considered to have PXS [74,106], whereas PXG is diagnosed when pseudoexfoliative material builds up in the trabecular meshwork, resulting in high intra-ocular pressures and subsequently glaucomatous optic nerve damage [106]. Individuals with PXS have a 5% risk of developing PXG after 5 years, a 15% risk after 10 years, and up to a 60% risk after 15 years [15,107], with PXG carrying a worse prognosis than POAG [108]. Moreover, PXS is a risk factor for developing acute angle closure glaucoma and cataract [10,108].

Although the majority of aqueous humour drains via the trabecular outflow pathway, a small percentage of aqueous drains through the uveoscleral pathway [102]. The latter allows aqueous to flow through the ciliary muscle, the supraciliary space, suprachoroidal space, and sclera, and into the lymphatic vessels in the orbit [102,109]. PXG has been characterised by reduced outflow of both the trabecular outflow pathway and the uveoscleral pathway [78,109]. Of note, in PXS, only uveoscleral outflow has been found to be significantly reduced [78,109]. The ciliary muscle influences uveoscleral outflow, the reduction of which is thought to be due to ciliary muscle ECM degradation by pseudoexfoliative deposits [78,109]. Prostaglandin analogues are a common IOP-lowering medication that increases uveoscleral outflow by remodelling ciliary muscle ECM [78]. Ex vivo studies investigating the mechanism of action of prostaglandin analogues have found ciliary muscle cells upregulate matrix metalloproteinase 2 and 3 [110]. Prostaglandin analogues have shown promising results in lowering IOP in PXS patients in short-term studies [111]; however, it is of note that patients with PXS and PXG are at high risk of loss of vision despite adequate IOP control [78].

Clinically, PXS eyes with pseudoexfoliative deposits showed increased TM pigmentation compared to controls [67]. Raised IOP in PXS is caused by mechanical obstruction and local destruction of TM by pseudoexfoliative material [67,108]. Both these mechanisms are thought to increase aqueous outflow resistance and therefore raise IOP [67,108].

Small nucleolar RNAs (snoRNAs) are non-coding RNAs that are involved in a variety of cellular processes [112,113]. SnoRNAs are being extensively researched due to their crucial role in post-transcriptional modification of various RNAs [113]. Their precise targeting capabilities give them the potential to be used as therapeutic targets and biomarkers in the treatment of different diseases [113]; however, this is still an emerging area in ocular research. In one study, seven snoRNAs: SNORD73B, SNORD58A, SNORD56, SNORA77, SNORA72, SNORA64, and SNORA32 have been identified in the aqueous humour of nine PXG patients and nine controls [112]. In the PXG group, five of the seven snoRNAs had significantly lower expression, while two had significantly higher expression, compared to the control group. This is a promising novel area of research for therapeutic targets for treatment of PXS and PXG.

### 2.9. Lens

The lens sits posterior to the iris and proximal to the ciliary body and zonules [114]. It is a clear structure that focuses light onto the retina [114]. The ciliary muscle controls zonular tension and allows the shape of the lens to change and focus on objects at different distances [77,114]. The lens is a curved structure composed of the nucleus at its core, which is surrounded by cortex and finally a thin lens capsule [114,115,116]. The lens capsule is a basement membrane primarily composed of collagen IV and laminin bound by nidogen and perlecan [116].

PXS is associated with increased risk of cataract formation, with nuclear sclerotic cataracts being the most common type [9,10]. Higher rates of lens subluxation, phacodonesis, zonular dialysis, and vitreous loss during cataract surgery have been reported in PXS patients [10].

Proteasome impairment occurs when the proteasome system is unable to properly degrade damaged, misfolded, or redundant proteins, leading to their unwanted buildup [5,117]. It is postulated that it plays a role in the pathogenesis of PXS [117]. Hayet et al. reported on a significant reduction in proteosome subunits in the lens capsules of PXS patients, an upregulation of endoplasmic reticulum stress markers such as DNAJB11, Caspase 12, and Synoviolin1, and upregulation of Heat Shock factor 1, a transcription factor that regulates proteosome activity [117]. Superoxide dismutase activity was found to be increased in PXS lens capsules, further suggesting the role of oxidative stress in its pathogenesis [118].

Terminal deoxynucleotidyl transferase-mediated dUTP-biotin nick end labelling (TUNEL) is a technique used to study apoptosis [119]. The lens epithelial cells of PXS patients were found to have significantly higher TUNEL immuno-positive cells and therefore higher rates of cellular death [119].

Numerous studies have reported on the physical properties of anterior lenses affected by pseudoexfoliation [76,120,121]. Mechanically, there were no significant differences between the anterior lens capsules of PXS and control groups [120]. However, Simsek et al. found PXS anterior lens capsules to be more elastic than controls [121] and Batur et al. reported that they were 18% thicker centrally [76].

## 3. Conclusions

In summary, although the pathogenesis of pseudoexfoliation syndrome is not fully understood, it is considered that abnormal ECM metabolism and function, oxidative stress, inflammation, and genetic and environmental factors are involved in an intricate interplay that causes this disease. *LOXL1*, *CACNA1A*, *CYP39A1*, *POMP*, *TMEM136*, *AGPAT1*, *RBMS3*, *SOD2*, *ALDH1A1*, and *SEMA6A* genes have been implicated in the pathogenesis of PXS. Chronic, low-grade inflammation and elevated markers of oxidative stress are observed in PXS, likely contributing to abnormal ECM turnover and fibrosis in ocular tissues.

We have discussed the many histopathological and structural consequences of PXS to structures within the ocular surface and anterior segment of the eye, which causes primary damage to tissue. Secondarily, the insoluble pseudoexfoliative material blocks the trabecular meshwork and increases intra-ocular pressure, which then causes glaucomatous optic neuropathy.

Given the potential for PXS to progress to PXG, understanding the epidemiology of PXS will be crucial for healthcare planning, early detection, and management. The global epidemiology of PXS highlights significant regional and ethnic variability, underscoring the complex interplay of genetic, environmental, and lifestyle factors in the development of the disease. As the global population ages, it is expected that the burden of PXS will increase, particularly in regions with higher levels of UV exposure and ageing populations. As vision loss significantly impacts quality of life, the prevalence of PXS-related glaucoma could impose a substantial public health burden in both developed and developing countries.

A deeper discussion could be made regarding healthcare costs and the long-term impact of PXS on healthcare systems, especially considering the direct costs associated with glaucoma management, as well as the indirect costs related to loss of productivity and reduced quality of life among patients. Total eye care costs are 27% higher in PXG patients than those with POAG due to a combination of additional clinic appointments, surgery, and higher rates of surgical complications [122,123]. Public health strategies aimed at screening for PXS and early detection of PXG could prove essential in mitigating the socio-economic impact of the syndrome.

As more data become available from diverse populations, a clearer understanding of how genetic, environmental, and lifestyle factors interact in the pathogenesis of PXS will help develop more targeted prevention and treatment strategies.

## 4. Future Directions

Considerable amounts of research into PXS and PXG have been conducted; however, we still do not fully understand its pathophysiology and the role of targeted therapy in managing this disease. Currently, traditional glaucoma medications (such as beta-blockers, prostaglandin analogues, and carbonic anhydrase inhibitors) used alone or in combination are prescribed to control IOP in patients with PXS and PXG [9]. As this population of patients has poorer response to these medications, laser and surgeries are often the next step to control IOP.

While there are currently no specific therapies aimed at reducing the accumulation of pseudoexfoliative material, several ongoing clinical and therapeutic research efforts focus on better understanding and managing the condition, particularly its progression to PXG [124,125,126]. Clinical trials are investigating the therapeutic benefit of early cataract surgery, selective laser trabeculoplasty, and minimally invasive glaucoma surgeries (MIGS) in PXG for IOP control. Clinical trials are investigating the outcomes of MIGS in PXS and PXG patients, with promising results showing that these surgeries may be effective in managing glaucoma while minimising risks associated with traditional trabeculectomy surgery [126].

Further exploration of the role of *LOXL1* and other genes associated with PXS is needed, particularly focusing on epigenetic mechanisms and gene–environment interactions. Exploration into gene therapies that could modulate *LOXL1* expression, or its protein function, could be the next step in developing drug therapies for PXS and PXG. As dysregulated cholesterol metabolism is associated with PXS, one area of research would be modulating the function of *CYP39A1*. There is ongoing research into gene editing techniques, such as CRISPR/Cas9, to potentially correct genetic mutations associated with eye diseases [127,128]. Although this research is still in its early stages, it holds the promise of providing a long-term solution by directly addressing the genetic causes of the condition.

Inflammation is another critical factor contributing to the progression of PXS. Clinical research is investigating the role of anti-inflammatory drugs, such as non-steroidal anti-inflammatory drugs (NSAIDs), in managing the ocular inflammation that accompanies PXS and PXG [129]. These treatments may help reduce the severity of glaucoma in PXG patients by preventing optic nerve damage.

Further research into the epidemiology of PXS, especially in underrepresented populations, will be key to developing targeted prevention and treatment strategies and improving our understanding of this increasingly prevalent condition. Expanding epidemiological studies to diverse populations will also help us to understand geographic and ethnic variations in PXS prevalence and genetic susceptibility.

Currently, there are no definitive, widely accepted biomarkers for PXS and there is still a need for further research to establish ones for clinical use, especially for early detection and monitoring of disease progression in PXS. Moreover, due to the systemic nature of PXS and its impact on extra-ocular organs such as the cardiovascular and central nervous system, it is crucial to develop validated imaging techniques for early detection of PXS and to develop treatments targeting the production and accumulation of pseudoexfoliative materials.

As these research efforts continue to unfold, they hold the potential to offer new hope for PXS and PXG patients by slowing disease progression, improving treatment efficacy, and ultimately preserving vision.

## Figures and Tables

**Figure 1 ijms-26-00532-f001:**
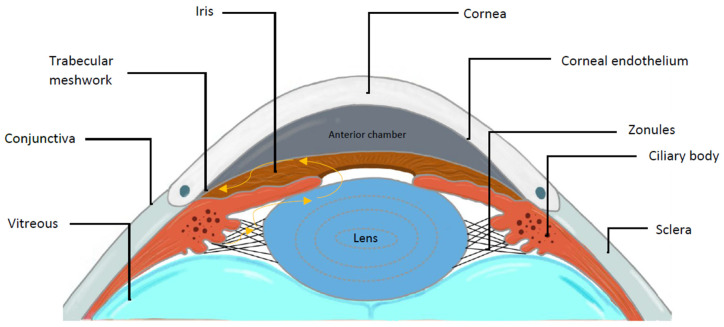
Labelled diagram of the anterior segment of the eye. The yellow arrows represent the flow of aqueous humour from where it is produced in the ciliary body to the trabecular meshwork.

**Figure 2 ijms-26-00532-f002:**
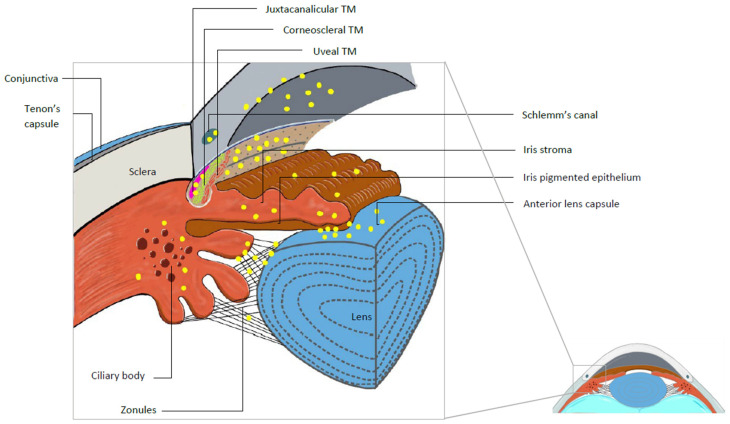
Labelled diagram of the pseudoexfoliation deposits (yellow dots) around the trabecular meshwork.

**Figure 3 ijms-26-00532-f003:**
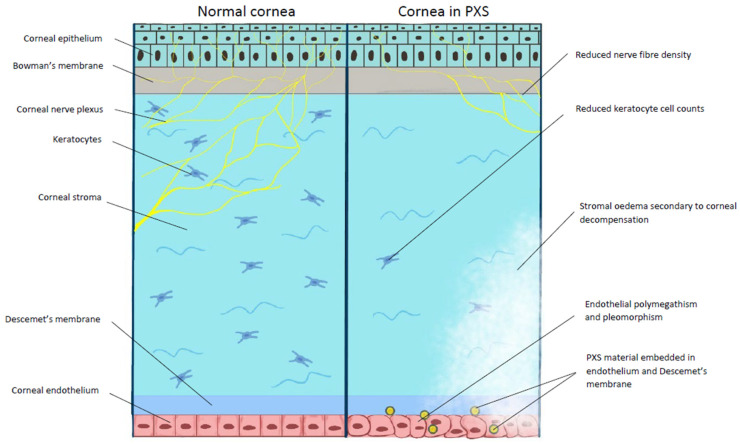
Diagram illustrating normal corneal anatomy (**left**) and structural changes caused by PXS material on the cornea (**right**).

**Figure 4 ijms-26-00532-f004:**
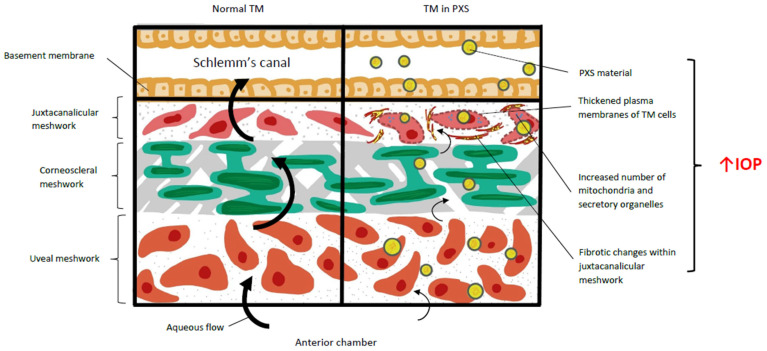
A labelled diagram illustrating effects of PXS deposits on the trabecular meshwork. The normal anatomy of the TM is depicted on the left. Structural changes to the TM caused by PXS are seen on the right.

**Figure 5 ijms-26-00532-f005:**
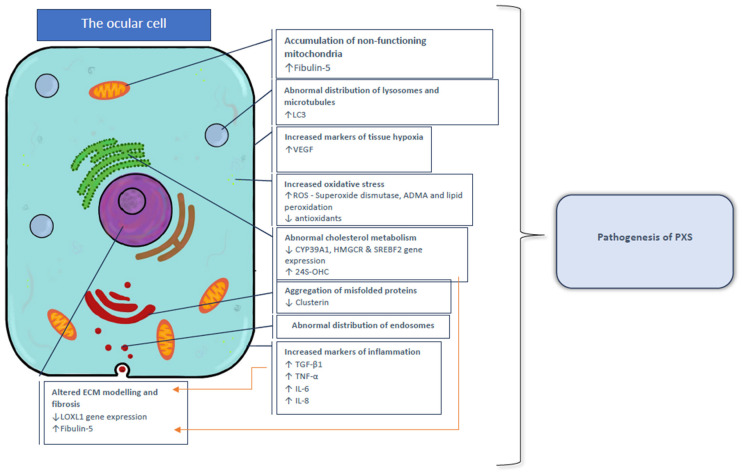
Summary of molecular mechanisms contributing to the pathogenesis of PXS to cells within the ocular surface and anterior segment of the eye.

## Data Availability

No new data were created or analysed in this study. Data sharing is not applicable to this article.

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
