# Peer review of "The Ocular Surface and the Anterior Segment of the Eye in the Pseudoexfoliation Syndrome: A Comprehensive Review"

_ijms, 2025, doi:10.3390/ijms26020532_

Round 1
Reviewer 1 Report
Comments and Suggestions for Authors
The article reviews in depth pseudoexfoliation syndrome (PXS), an age-related fibrillopathy characterized by the accumulation of insoluble fibrillar material in the eye and extraocular tissues.The authors present a comprehensive review of the current literature and a comprehensive assessment of the effects of PXS on ocular structures, integrating genetic, molecular and clinical knowledge. Specifically, the review analyzes a wide range of studies, generating a comprehensive review of genetic (e.g. LOXL1 and CYP39A1), bio-molecular (primarily inflammation and oxidative stress) data. The involvement of specific genes and their variants (e.g. LOXL1, CACNA1A) is well documented, providing a robust basis to understand the genetic predisposition to PXS. It is specifically analyzed how LOXL1 regulates collagen and elastin cross-linking, as well as how some polymorphisms (e.g. rs1048661) show significant associations with the disease. Other genes involved are also reported such as CYP39A1 (cholesterol metabolism), CACNA1A (calcium function), and SOD2 (oxidative stress). The clinical and instrumental alterations are then analyzed and described, describing how the deposits of pseudoexfoliative material compromise the production and stability of the tear film, causing dry eye, alterations affecting the Tenon's capsule and zonular fibers, corneal alterations ranging from reduced endothelial cell density to pleomorphism and alterations of the sub-basal nerve and other ocular structures such as the iris, ciliary body and trabecular meshwork. This description is excellently performed, reporting in detail the effects of PXS on each component of the anterior segment. The interconnection between the pseudoexfoliative deposits and the increase in intraocular pressure (primary cause of pseudoexfoliative glaucoma – PXG) is also underlined and highlighted. Finally, the main environmental risk factors are evaluated and described, such as oxidative stress, UV exposure, excessive coffee consumption and folate deficiency. Secondly, the association of PXS and systemic diseases (such as Alzheimer's and arterial hypertension) is correlated and analyzed, underlining the multifactorial role of the syndrome. In the last final part, epigenetic interactions, diagnosis and current future perspectives are described, which mainly focus on an increasingly early diagnosis through advanced imaging and the development of targeted therapies against pseudoexfoliative material. Despite the undisputed value of the comprehensive and didactic review, the article does not present original or experimental data, therefore not bringing significant innovations to the currently existing literature. Furthermore, although they are cited, the discussion about the global epidemiology of PXS and the socio-health implications could have been deepened and expanded. Finally, in the last part, despite the lack of specific therapies to reduce the accumulation material being clearly mentioned, current clinical-therapeutic research and trials could have been cited and discussed in more depth. The language of the study is appropriate, clear and concise, didactic and particularly suitable for a specialist audience. However, it could be difficult to access for a non-specialist audience. In conclusion, the article represents an excellent contribution to the understanding and description of pseudoexfoliation syndrome, providing a solid basis for further future studies. The strength of this article therefore lies in its ability to offer a complete and well-structured picture of the effects of PXS on the eyes and the body. To further improve this article, it would be useful to present original data or comparative analyses to strengthen the conclusions, describe more carefully the epidemiological data and the implications of this pathology from an environmental and socio-economic point of view, provide simplified summaries to facilitate understanding even for non-specialists.
Author Response
We would like to thank the reviewer for taking the time to review our manuscript. We have addressed comments individually and listed specific changes made to the manuscript in the form of a table below. Corresponding corrections were highlighted in the re-submitted version of the paper.
|
Reviewer summary |
||
|
The article represents an excellent contribution to the understanding and description of pseudoexfoliation syndrome, providing a solid basis for further future studies. The strength of this article therefore lies in its ability to offer a complete and well-structured picture of the effects of PXS on the eyes and the body. To further improve this article, it would be useful to present original data or comparative analyses to strengthen the conclusions, describe more carefully the epidemiological data and the implications of this pathology from an environmental and socio-economic point of view, provide simplified summaries to facilitate understanding even for non-specialists. |
||
|
Reviewer comment |
Response |
Change |
|
Despite the undisputed value of the comprehensive and didactic review, the article does not present original or experimental data, therefore not bringing significant innovations to the currently existing literature. |
Thank you for your thoughtful feedback. While the article does not present original experimental data, the primary goal was to provide a comprehensive and didactic review of the existing literature, highlighting key trends, insights, and areas for future research. We believe this synthesis of current knowledge offers valuable perspectives that can guide further exploration in the field. We agree that incorporating original data would be a valuable contribution and could form the basis for future work. |
N/A |
|
Furthermore, although they are cited, the discussion about the global epidemiology of PXS and the socio-health implications could have been deepened and expanded. |
We thank the reviewer for this comment. We have added a longer discussion about the epidemiology of PXS and the socio-health implications in our revised manuscript. |
Page 3, paragraph 1 and 2 Page 19 paragraph 5 Page 20 paragraph 1-3 |
|
Finally, in the last part, despite the lack of specific therapies to reduce the accumulation material being clearly mentioned, current clinical-therapeutic research and trials could have been cited and discussed in more depth. |
Thank you for your valuable suggestion. We have revised the last section to include a more in-depth discussion of current clinical-therapeutic research and trials, as well as relevant citations. We hope these additions address your concern and enhance the article. |
Page 20 paragraph 4-5 Page 21 paragraph 1-3 |
|
Provide simplified summaries to facilitate understanding even for non-specialists.
|
Thank you for the suggestion. To facilitate understanding for non-specialists, we have added three new original illustrated diagrams that summarize the anatomical changes in the cornea, trabecular meshwork, and ocular cells in PXS. Additionally, we have included a table summarizing the aqueous humour components with altered concentrations in PXS. We hope these additions help clarify the content for a broader audience. |
Page 9, figure 3 Page 15, table 1 Page 15, figure 4 Page 16, figure 5 |
Reviewer 2 Report
Comments and Suggestions for Authors
Thomas et al. summarized the ocular surface and the anterior segment of the eye in the pseudoexfoliation syndrome, as a comprehensive review paper. Its topic is interesting and needs to be summarized.
Depending on the tissue layers, information is described. It might be better to make a table to show it easily for the future reader.
In each tissue/cell type, pathologic signaling cascades exist. Its aspect has not been discussed. Drawing a schematic illustration might be helpful.
Listed (LOXL1, CACNA1A, CYP39A1, POMP, TMEM136, AGPAT1, RBMS3, SOD2, ALDH1A1 and SEMA6A) genes could have some connections. Would it be available to make an interaction analysis? You may check STRING if available.
How about the metabolites in each tissue? Is there any biomarker to be detectable? It should be discussed.
There are some typos. It is a minor point to check those and revise them.
Author Response
We would like to thank the reviewer for taking the time to review our manuscript. We have addressed comments individually and listed specific changes made to the manuscript in the form of a table below. Corresponding corrections were highlighted in the re-submitted version of the paper.
|
Reviewer summary |
||
|
The reviewer finds the paper on pseudoexfoliation syndrome (PXS) interesting but suggests summarising the information more clearly, possibly with a table for easier reference. They recommend discussing pathological signaling cascades in tissues and including a schematic illustration. Additionally, they suggest exploring gene interactions (e.g., LOXL1, CACNA1A) through STRING analysis and discussing potential metabolites or biomarkers in PXS. Finally, they note some minor typographical errors that should be corrected. |
||
|
Reviewer comment |
Response |
Change |
|
Depending on the tissue layers, information is described. It might be better to make a table to show it easily for the future reader.
|
Thank you for the suggestion. To facilitate understanding for the future reader, we have added three new original illustrated diagrams that summarize the anatomical changes in the cornea, trabecular meshwork, and ocular cells in pseudoexfoliation. Additionally, we have included a table summarizing the aqueous humour components with altered concentrations in pseudoexfoliation. I hope these additions help clarify the content for a broader audience. |
Page 9, figure 3 Page 15, table 1 Page 15, figure 4 Page 16, figure 5 |
|
In each tissue/cell type, pathologic signaling cascades exist. Its aspect has not been discussed. Drawing a schematic illustration might be helpful.
|
Thank you for your insightful comment. While the presence of pathological signaling cascades in each tissue/cell type is acknowledged, the literature regarding the specific signaling pathways involved in pseudoexfoliation (PXS) remains uncertain, with various sources offering contradictory information. As such, we chose not to include a detailed discussion or schematic illustration of these cascades to avoid presenting speculative or inconsistent data. We appreciate your understanding. |
N/A |
|
Listed (LOXL1, CACNA1A, CYP39A1, POMP, TMEM136, AGPAT1, RBMS3, SOD2, ALDH1A1 and SEMA6A) genes could have some connections. Would it be available to make an interaction analysis? You may check STRING if available.
|
Thank you for the suggestion. We reviewed the potential gene interactions using STRING but found only a minor link between two of the smaller, less significant genes, with no current associations identified for the others. This highlights the need for further research to better understand the connections between these genes. We appreciate your insight on this matter. |
N/A |
|
How about the metabolites in each tissue? Is there any biomarker to be detectable? It should be discussed |
Thank you for your suggestion. We have added a paragraph addressing the current lack of definitive biomarkers for PXS, including metabolites in different tissues. This remains an area of active research, and identifying such biomarkers will be crucial for future advancements in diagnosis and treatment. |
Page 21 paragraph 4 |
|
There are some typos. It is a minor point to check those and revise them. |
Thank you for your comment, we have gone through the manuscript to edit these typos. |
|